# Anti-Inflammatory Drugs in Patients with Ischemic Heart Disease

**DOI:** 10.3390/jcm10132835

**Published:** 2021-06-27

**Authors:** Ana María Pello Lázaro, Luis M. Blanco-Colio, Juan Antonio Franco Peláez, José Tuñón

**Affiliations:** 1Department of Cardiology, IIS-Fundación Jiménez Díaz, 28040 Madrid, Spain; ampello@quironsalud.es (A.M.P.L.); jafranco@quironsalud.es (J.A.F.P.); 2Department of Medicine, Autónoma University, 28029 Madrid, Spain; 3Laboratory of Vascular Pathology, IIS-Fundación Jiménez Díaz, 28040 Madrid, Spain; lblanco@fjd.es; 4CIBERCV, 28029 Madrid, Spain

**Keywords:** inflammation, atherosclerosis, coronary heart disease, biomarkers

## Abstract

Inflammation has long been known to play a role in atherogenesis and plaque complication, as well as in some drugs used in therapy for atherosclerotic disease, such as statins, acetylsalicylic acid, and modulators of the renin-angiotensin system, which also have anti-inflammatory effects. Furthermore, inflammatory biomarkers have been demonstrated to predict the incidence of cardiovascular events. In spite of this, and with the exception of acetylsalicylic acid, non-steroidal anti-inflammatory drugs are unable to decrease the incidence of cardiovascular events and may even be harmful to the cardiovascular system. In recent years, other anti-inflammatory drugs, such as canakinumab and colchicine, have shown an ability to reduce the incidence of cardiovascular events in secondary prevention. Colchicine could be a potential candidate for use in clinical practice given its safety and low price, although the results of temporary studies require confirmation in large randomized clinical trials. In this paper, we discuss the evidence linking inflammation with atherosclerosis and review the results from various clinical trials performed with anti-inflammatory drugs. We also discuss the potential use of these drugs in routine clinical settings.

## 1. Introduction

Current evidence places atherosclerosis as a chronic condition in which inflammation plays a primary role [1]. In spite of this, anti-inflammatory drugs have long been said to increase the incidence of cardiovascular events, and only recently have some anti-inflammatory drugs been shown to decrease the development of cardiovascular events. In this article, we review the evidence linking inflammation to atherosclerosis and assess the effects of anti-inflammatory drugs on the progression of this disorder.

### 1.1. Atheroma Plaque Formation and Its Inflammatory Components

Atherosclerotic lesions appear more frequently in areas with low shear stress, leading to an increased turnover of endothelial cells [2]. This makes the arterial wall more permeable, allowing the entry of circulating molecules [3]. Furthermore, atherogenesis exhibits the different components of inflammation, including recruitment of inflammatory cells, cell proliferation, sclerosis, and neovascularization.

Endothelial dysfunction is the initial event in atherogenesis [1] and is triggered by lower availability of nitric oxide [4,5,6]. In endothelial dysfunction, there is increased vascular permeability, allowing an excess of low-density lipoprotein (LDL) particles to enter the vessel wall, which exceeds the capacity of the cholesterol reverse transport system to remove this LDL through the wall. Thus, these particles remain in the sub-endothelial space for longer, where they undergo mild oxidation and evolve into minimally modified LDL particles [7]. This may activate the expression of pro-inflammatory transcription factors such as nuclear factor-κB (NF-κB) [8,9], favoring the expression of adhesion molecules and chemoattractant proteins [8]. In this way, monocytes adhere to endothelial cells and then may enter the vascular wall as they are stimulated by chemokines, such as monocyte chemoattractant protein-1 (MCP-1) [10,11]. Monocytes in the sub-intimal space turn into macrophages [12], which phagocytize oxidized LDL, finally evolving into foam cells. These cells die, releasing several toxic substances that injure the endothelium [13].

This situation favors a pro-thrombotic state and platelet adhesion to the vascular wall with the consequent release of different growth factors and stimulation of cytokine production, which perpetuates the inflammatory response [14]. These growth factors induce proliferation and migration of smooth muscle cells from the media to the intima [15,16]. These cells synthetize matrix proteins such as collagen and elastin to form the fibrous cap [17,18] in a process that resembles the sclerosis reaction in an inflammatory process. Moreover, angiogenic factors induce microvessel formation [19], which may favor the recruitment of more inflammatory cells [20].

### 1.2. Relationship between Pro-Inflammatory States and Acute Coronary Events

In addition to its role in atherogenesis, inflammation also promotes plaque complication. Most acute coronary syndromes (ACS) are due to fibrous cap rupture [21], in which the inflammatory response plays a key role [22]. Plaques responsible for ACS display higher NF-κB activity [23,24] and more frequent infiltration by inflammatory cells than stable ones [25,26]. Inflammatory cells degrade and weaken the extracellular matrix by producing enzymes such as metalloproteinases (MMPs) [17,27], while there is also a shortening of collagen synthesis [27]. The above favors rupturing of the fibrous cap of atheroma, allowing contact between blood and the lipid core, which is rich in tissue factor, and thus triggers thrombosis [21].

Several studies of patients with ACS have demonstrated not only increased plasma levels of inflammatory molecules (MCP-1, adhesion molecules, MMPs, or CD40 ligand (CD40L)), but also enhanced NF-κB activity in circulating leukocytes [28,29]. Recently, it has been shown that higher MCP-1 levels in human endarterectomy atheroclerosis plaques are associated with histopathologic features of plaque vulnerability (low collagen content, large lipid core, low smooth muscle cell burden, high macrophage burden and intraplaque hemorrhage) and these findings increase the risk of ACS [30]. Moreover, MCP-1 has shown prognostic value, since higher levels are associated with an increased risk of cardiovascular events in patients with both acute [31] and chronic coronary artery disease (CAD) [32]. Plasma levels of MCP-1 predict cardiovascular events in patients with stable CAD and persistent inflammation, defined as C-reactive protein (CRP) levels above 2 mg/L as measured by high-sensitivity methods [33]. Not only plasma levels of MCP-1 have prognostic value, but MCP-1 plaque levels are also associated with a higher risk of major adverse vascular events and strokes in the first 30 days after endarterectomy [30].

Furthermore, it is important to note the link between inflammation and thrombosis. For instance, NF-κB acts by up-regulating the expression of tissue factor, which is present in the lipid core of atherosclerotic plaque and is key to initiating the thrombosis that triggers acute coronary events [13]. Additionally, CD40L, which triggers the inflammatory cascade, has been found to play a role in thrombus stability [34].

The close temporal and causal link between inflammation and ACS has also come to the forefront in other settings such as in research surrounding COVID-19. In patients with simultaneous ST-elevation ACS and COVID-19, systemic inflammatory response was significantly higher than in patients without COVID-19, including higher levels of D-dimer, troponin-T, fibrinogen, ferritin, and CRP, as well as lower lymphocyte counts [35]. Moreover, patients with ST-elevation ACS and COVID-19 have a higher rate of thrombotic complications, with higher thrombotic burden, multivessel disease and stent thrombosis, and worse myocardial reperfusion degree [36].

### 1.3. Inflammatory Biomarkers and Atherosclerosis

Given the role of inflammation in atherosclerosis, it is expected that inflammatory mediators will be released into circulation and serve as biomarkers to predict cardiovascular risk. Among the biomarkers reported to date, plasma CRP is the most widely studied indicator of vascular inflammation, as it identifies low but persistent levels of inflammation. Nevertheless, the increase in CRP levels is non-specific, as they rise in any inflammatory context, and results on its predictive value have been contradictory [37,38]. The European Society of Cardiology guidelines for cardiovascular prevention do not advise routine assessment of CRP levels as part of risk assessment [39], while the American College of Cardiology guidelines suggest considering use of CRP levels if a risk-based treatment decision is uncertain after quantitative risk measurement [40].

Several biomarkers have been associated with inflammation and are recognized as potential tools with which to monitor the progression of atherosclerosis. High MCP-1 plasma levels are present in patients with CAD [41], and provide independent prognostic value for this disorder [31,32,33]. Recently published studies have stated that higher plasma MCP-1 levels are associated with higher cardiovascular mortality, even in individuals without established cardiovascular disease, which is demonstrative of the role of MCP-1-signaling in atherosclerosis [42].

Substantial attention has been paid to lipoprotein-associated phospholipase A_2_ (Lp-PLA_2_) [43], an enzyme highly expressed by macrophages and one that is closely involved in the inflammatory response associated with plaque rupture [44,45]. Circulating Lp-PLA_2_ levels have been associated with the risk of cardiovascular events [46,47]. However, Lp-PLA2 is not a fully validated biomarker for clinical practice.

Other inflammatory biomarkers have also been related to cardiovascular disease. The adhesion molecules expressed by the endothelium, such as intercellular adhesion molecule-1 (ICAM-1) and vascular cell adhesion molecule-1 (VCAM-1), are widely accepted biomarkers of endothelial dysfunction [48]. Several studies have associated circulating levels of ICAM-1, VCAM-1, and E-selectin with the severity and complications of CAD [49,50]. However, cell adhesion molecules are also expressed by inflammatory cells and platelets, and they have limited diagnostic value when measured alone [51].

Among other potential biomarkers, circulating levels of tumor necrosis factor-like weak inducer of apoptosis (TWEAK) are a potential biomarker. TWEAK levels in plasma are associated with an increased cardiovascular risk [52], although their prognostic value has not been confirmed [32]. In addition, circulating levels of several interleukins (IL) such as IL-6, IL8, and IL-18 are associated with the presence of acute and chronic CAD [53,54,55] and may even have prognostic value [56].

As we have seen here, many circulating proteins related to inflammation have been shown to be associated with the presence of atherosclerosis and may be of prognostic value. However, data have not been consistent enough so as to advise routine use of these biomarkers in clinical practice.

## 2. Drugs Used in Therapy for Atherosclerotic Disease Reduce Inflammation

Several drugs with proven efficacy in atherosclerotic disease have demonstrated anti-inflammatory effects.

Acetylsalicylic acid inhibits NF-κB activation by reducing expression of some adhesion molecules such as VCAM-1 and E-selectin, which are key to atherosclerotic and inflammatory processes, as they allow monocytes to adhere to endothelial cells [57] with a dose-dependent effect.

The clinical practice guidelines of the European Society of Cardiology recommend using renin-angiotensin system modulators in ischemic heart disease patients with heart failure, hypertension, or diabetes (class IA), and even in those with chronic CAD who are at very high cardiovascular risk (class IIa) [58,59]. These drugs improve endothelial function, and we have found that they inhibit NF-kB activation, the expression of MCP-1, and macrophage infiltration in a rabbit model of atherosclerosis [60]. This is consistent with evidence of over-expression of angiotensin-converting enzyme in atherosclerotic lesions [61].

The deposit of lipids into the arterial wall is a trigger of inflammatory response in atherogenesis. Thus, LDL storage inside the vascular wall favors the expression of adhesion molecules, cytokines, and other pro-inflammatory molecules such as CD40L and NF-κβ, which activate the inflammatory response [62]. Statins, the most frequently prescribed lipid-lowering drugs, reduce cardiovascular events in primary and secondary prevention [63,64]. We have demonstrated that they decrease NF-kB activation and MCP-1 expression and reduce the macrophage infiltrate in experimental atherosclerosis [65], and many studies have confirmed these anti-inflammatory effects of statins and other lipid-lowering drugs [66].

In sum, successful drugs used in therapy for atherosclerosis also have anti-inflammatory properties. However, this does not prove that their clinical benefits are the result of these effects, as they also have other beneficial actions. For instance, acetylsalicylic acid has anti-thrombotic effects and RAS modulators both improve endothelial function and decrease blood pressure. In the case of lipid therapy, it has been suggested that statins could have anti-inflammatory actions that are independent of their lipid-lowering ability. However, these effects have been demonstrated mainly in in-vitro experiments and animal models. The clinical benefit of these drugs is proportional to the LDL decrease, and it seems that, in humans, they do not reach the concentrations necessary to reduce inflammation in atherosclerotic plaques [62].

## 3. Classical Anti-Inflammatory Drugs

Given the role of inflammation in atherosclerosis, anti-inflammatory drugs can be expected to reduce the incidence of cardiovascular events. However, this has not been confirmed in clinical studies.

In addition to the failure of steroids to improve the prognosis of patients with unstable angina [67], non-steroidal anti-inflammatory drugs (NSAID) have even been shown to increase the incidence of cardiovascular events in multiple studies [68] with the known exception of acetylsalicylic acid [69]. Small studies published in the 1980s showed the adverse effect of indomethacin (oral or intravenous) in patients with coronary ischemic disease due to a vasoconstrictor effect with reduced coronary blood flow and increased coronary resistance [70,71]. This paradox likely reflects the complexity of the cyclooxygenase (COX) pathway inhibited by these drugs. COX metabolizes arachidonic acid into prostaglandin (PG) G_2_ (Figure 1). The final products are PGs, which may have a vasodilatory effect, such as PGI_2_, but others may have pro-inflammatory (PGE_2_) and pro-thrombotic (Thromboxane A_2_) actions [69].

There are two known isoforms of COX: COX-1 and COX-2. Although their product is the same, it has been said that COX-1 is co-expressed with PG synthases that lead to the formation of protective PGs, while COX-2 is expressed along with PG synthases that promote the formation of pro-inflammatory PGs [72]. In accordance with this idea, COX-1 is a constitutive enzyme that is expressed in several tissues and acts by regulating homeostasis and maintaining the integrity of gastrointestinal mucosa and kidney vascular flow. Conversely, COX-2 is an inducible enzyme present mainly in inflamed tissues, and is likely to mediate the synthesis of pro-inflammatory PGs. This idea suggests that the anti-inflammatory effects could be derived from selective COX2 inhibition, avoiding gastrointestinal toxicity and renal damage mediated by COX-1 antagonism [73].

However, findings from clinical trials do not support this initial hypothesis. The VIGOR trial demonstrated an increased incidence of thrombotic cardiovascular events in more than 8000 patients with rheumatoid arthritis [74]. This could be explained by either a higher pro-thrombotic effect of rofecoxib or by a greater anti-thrombotic effect of naproxen. In order to answer this question, a meta-analysis including more than 28,000 patients compared rofecoxib against placebo, naproxen, and other non-selective NSAIDs, demonstrating a higher rate of cardiovascular events in the rofecoxib group only upon comparison with naproxen, but there were no significant differences when comparing rofecoxib with placebo or other non-selective NSAIDs [75].

In the CLASS trial, however, which included more than 8000 patients with arthritis, patients treated with celecoxib showed no differences in the incidence of thrombotic cardiovascular events as compared to those receiving ibuprofen or diclofenac [76]. It should be noted here that in this trial the use of acetylsalicylic acid was allowed. In contrast, acetylsalicylic acid treatment was an exclusion criterion in the VIGOR study, although patients with cardiovascular disease were included [74]. Furthermore, the true increase of cardiovascular events in the celecoxib group could be masked by two other factors: first, diclofenac and ibuprofen have a lesser antiplatelet effect than naproxen [75], and second, diclofenac has higher COX-2 inhibition than naproxen with a higher pro-thrombotic effect due to lower activity of PG-I2 [77].

Nevertheless, a meta-analysis with 30 case-control and 21 cohort studies (more than 2.7 million individuals) found rofecoxib (HR 1.45; 95% CI 1.33–1.49), diclofenac (HR 1.40; 95% CI 1.27–1.55), and celecoxib (HR 1.17; 95% CI 1.08–1.27) to be associated with the highest risk of cardiovascular events, even at low doses, while ibuprofen only increased the risk at higher doses. Naproxen had a neutral effect at any dose, and its cardiovascular risk profile was favorable even when compared to ibuprofen (relative risk reduction 0.92; 95% CI 0.87–0.99) [68]. The increased cardiovascular risk of NSAIDs has also been demonstrated with real-world data. A cohort of 446,763 individuals (61,460 with acute myocardial infarction) confirmed an increased risk of acute myocardial infarction in NSAIDs users (including naproxen). Risk was also greatest during the first 30 days and with higher doses [78].

These data differ from the results of PRECISION trial comparing celecoxib (200 mg per day), naproxen (750 mg per day), and ibuprofen (1800 mg per day) which studied 24,081 patients with osteoarthritis or rheumatoid arthritis and established cardiovascular disease or high cardiovascular risk. Celecoxib reached the non-inferiority hypothesis of the study with respect to naproxen and ibuprofen regarding the incidence of major cardiovascular events, and was associated with less adverse renal and gastrointestinal events [69].

In order to understand these clinical results, we must take into account two important features. First, COX-2 is not an exclusive mediator of the synthesis of pro-inflammatory PGs. In smooth muscle cells from rabbits, COX-2 has been shown to mediate HDL-induced PG-I_2_ synthesis [79]. Second, COX-1 does not only promote the synthesis of PGs that induce vasodilatation; rather, it mediates TXA1 synthesis in platelets [80]. Then, COX-1 blockade may be beneficial from a cardiovascular point of view.

Thus, while acetylsalicylic acid protects against cardiovascular events, the remaining NSAIDs, including specific and non-specific COX-2 inhibitors, show an adverse or, in some studies, neutral effect. We believe that this does not constitute evidence against the role of inflammation in atherosclerosis, but only demonstrates the great complexity of the arachidonic acid pathway. In accordance with this, anti-inflammatory therapies blocking other pathways have been successful in reducing the incidence of cardiovascular events.

## 4. Non-Classical Anti-Inflammatory Drugs

The CANTOS (Canakinumab Anti-inflammatory Thrombosis Outcome Study) was the first large-scale trial to demonstrate that canakinumab, a monoclonal antibody that blocks interleukin-1β (IL-1β), can reduce the incidence of cardiovascular events. More than 10,000 patients with previous myocardial infarction and CRP levels > 2 mg/L, were randomized to receive placebo or three different subcutaneous doses of canakinumab every 3 months (50 mg, 150 mg, and 300 mg). After a median follow-up of 3.7 years, patients randomized to 150 mg of canakinumab had a 15% reduction in the primary endpoint (myocardial infarction, non-fatal stroke, or cardiovascular death) without changes in plasma lipid levels, except for a slight increase in triglyceride levels [81]. Risk reduction was even higher in patients with an above-average decrease in CRP levels (<2 mg/L) [82]. A lesser incidence of lung cancer, cancer-related death, arthritis, and gout was observed in the active group, though this group had a higher incidence of fatal infections, sepsis, neutropenia, and thrombopenia, without excess bleeding. These adverse effects, along with the high cost of the drug and lack of other trials confirming its clinical benefit, have led the European Medicines Agency to prepare a report that advises against the use of this drug, considering that the benefit –risk balance is currently negative for secondary prevention of major adverse cardiovascular events in patients following a myocardial infarction [83]. In addition, although 93% of patients were taking lipid-lowering drugs, average LDL level was 82.4 mg/dL, which is well above the target of 55 mg/dL recommended at present for very high-risk patients [84]. Given that lipid-lowering drugs may interfere with this pathway by decreasing IL-1β expression (Figure 2) [85,86,87] it would be interesting to know if canakinumab is really capable of adding benefit to patients reaching this target, or whether intensive lipid-lowering therapy is sufficient.

Another anti-inflammatory drug that has attracted interest in the cardiovascular area is methotrexate. This drug is used in the treatment of rheumatologic and oncologic disorders, and it may reduce CRP, IL-6, and TNF-α levels. Furthermore, a meta-analysis showed a decrease in cardiovascular events when comparing methotrexate to other treatments for rheumatologic disorders [88]. The CIRT (Cardiovascular Inflammation Reduction Trial) trial was designed to confirm these results. In the trial, 4786 patients with previous myocardial infarction, multivessel coronary disease, diabetes, or metabolic syndrome were randomized to receive either low-dose methotrexate (15–20 mg weekly) or placebo. CRP levels were not considered an inclusion criterion (Table 1), although diabetes or metabolic syndrome could result in a certain degree of residual chronic inflammation. After a median follow-up of 2.3 years, methotrexate failed to decrease the incidence of the primary endpoint (myocardial infarction or non-fatal stroke or cardiovascular death) and did not reduce CRP, IL-6, or IL-1β plasma levels. Despite a lower percentage of patients receiving lipid-lowering therapy (85.9%) as compared with CANTOS, average LDL levels were lower (68 mg/dL) than those of patients in the CANTOS trial. In addition, a higher rate of adverse events was observed in the methotrexate group, mainly driven by an increased rate of liver enzymes, leukopenia, anemia, and non-basal cell skin cancer [89].

We cannot rule out the possibility that the lack of effect observed in the CIRT trial was due to an absence of high CRP levels as an inclusion criterion. Also, the low LDL levels among the patients in this trial may leave no margin for an additional benefit of methotrexate. However, an interesting possibility would be that the cardiovascular benefit of anti-inflammatory drugs depends on the pathway targeted. In this regard, other anti-inflammatory drugs with different mechanisms of action have shown neutral effects on cardiovascular outcomes [94,95].

Perhaps the most attractive anti-inflammatory drug for cardiovascular risk reduction is colchicine, a powerful, cheap, and orally administered anti-inflammatory drug that has been known for centuries. Colchicine has demonstrated efficacy in gout, pericarditis, and familial Mediterranean fever [96]. Its mechanism of action is different from the COX and IL-1β pathways, as it inhibits microtubule formation, neutrophil chemotaxis, adhesion, and activation, in addition to the interaction between neutrophils and platelets. These actions may avoid instability, rupture of the atheroma plaque, and progression of atherosclerotic disease [97].

During the last decade, several studies have attempted to confirm whether colchicine may improve cardiovascular outcomes (Table 1). The first study addressing the efficacy of colchicine in atherosclerosis was the LoDoCo (Low-Dose Colchicine for secondary prevention of cardiovascular disease) trial, published in 2013 [90]. Patients with stable ischemic heart disease were randomized to 0.5 mg of colchicine daily or conventional treatment over a median follow-up of 3 years. Colchicine treatment was associated with a significant reduction in primary combined endpoint (ACS, out-of-hospital resuscitated cardiac arrest, or non-cardioembolic ischemic stroke) with differences mainly driven by a lower rate of ACS. These results were maintained when patients who did not tolerate gastrointestinal side effects were excluded [90]. Despite several limitations, including small sample size (532 patients), an open-label design, and this not being a placebo-controlled trial, these were promising results that led to the design of large clinical trials to confirm its findings.

In 2020, the LoDoCo-2 trial was published (Table 1) [91]. In this trial, 5522 patients with stable ischemic heart disease were randomized to receive 0.5 mg daily of colchicine or placebo. After a median 28.6-month follow-up, treatment with colchicine was associated with a 31% reduction in the incidence of the primary endpoint (cardiovascular death, spontaneous myocardial infarction, ischemic stroke, or ischemia-driven coronary revascularization) as well as lower gout incidence.

Colchicine has also been tested in the context of ACS when the risk of recurrent cardiovascular events is higher and when inflammation is very likely to play a prominent role. Thus, the COLCOT (Efficacy and Safety of Low-Dose Colchicine after Myocardial Infarction) trial compared colchicine to a placebo in 4745 patients with a recent myocardial infarction, with a median follow-up of 22.6 months. Patients receiving colchicine showed a significant reduction in the incidence of the primary endpoint (death from cardiovascular causes, resuscitated cardiac arrest, myocardial infarction, stroke, or urgent hospitalization for angina leading to coronary revascularization) which was mainly driven by a lower incidence of stroke and urgent hospitalizations for angina. There were no differences in overall side effects, although when analyzed separately, the colchicine group had a slightly higher incidence of gastrointestinal events (nausea or flatulence) and pneumonia, without differences in septic shock, cancer, myopathy, or pancytopenia [92]. The magnitude of benefit with colchicine in this study is at least in the same range as canakinumab in the CANTOS trial. Unlike the CANTOS trial, however, high CRP levels were not among the inclusion criteria. Thus, only a small subgroup of COLCOT patients had CRP level assessed (207 patients) with an average of 4.28 mg/L and with a reduction of more than 65% at 6 months, with no differences found between treatment groups.

A few months later, the Australian Colchicine in Patients with Acute Coronary Syndrome (COPS) study compared colchicine versus placebo in 795 patients with ACS, and evidence of CAD managed with percutaneous coronary intervention or medical treatment [93]. After 12 months of follow-up, there were no differences in the primary outcome (all-cause mortality, ACS, ischemia-driven urgent revascularization, and non-cardioembolic ischemic stroke). Additionally, the rate of total mortality was higher in the colchicine group (HR 8.20; 95% CI 1.03–65.61; *p* = 0.047), especially non-cardiovascular death (5 versus 0; *p* = 0.024, log-rank). The adverse effects were similar in both groups. The difference between these results and those of the COLCOT trial could have several explanations. First, all patients were randomized during hospitalization while in COLCOT the median interval from ACS to inclusion was 14 days. It is well-known that the early phase after an ACS carries the highest risk of adverse events. Second, the dose of colchicine was highest during the first month, and this may have contributed to the negative results. Finally, the higher rates of non-cardiovascular deaths may reflect a type 1 error because of the small sample and few events for analysis.

Recently, a meta-analysis including these four trials and another one with a smaller sample size was published. After analyzing data from 11,816 patients, colchicine treatment showed a significant reduction in the primary efficacy endpoint of cardiovascular death, myocardial infarction, or stroke (RR 0.75; 95% CI 0.61–0.92; *p* = 0.005), which was mainly driven by a lower incidence of myocardial infarction and stroke. There were no differences in all-cause death or adverse events [98].

In conclusion, given these results, and taking into account that colchicine is a safe and cheap drug with a well-known mechanism of action, this drug could be of use in treating patients as part of secondary prevention. Moreover, the absence of high CRP in the inclusion criteria of these studies makes its use easier. Nevertheless, it would be interesting to analyze whether patients with a high inflammatory state derive more benefit from colchicine.

## 5. Present and Future Role of Anti-Inflammatory Drugs in Atherosclerosis

Although it is clear that cardiovascular risk factors such as dyslipidemia may promote inflammation, the degree of inflammation may be different among patients due to genetic factors. The role of genetics in the development of the inflammatory response that triggers atherosclerosis has been investigated for decades. For instance, several genes related to disturbances in lipid and lipoproteins have been associated with an increased risk of ischemic heart disease [99]. Furthermore, genetic factors might explain why patients without risk factors develop cardiovascular diseases [100]. In the population-based Bruneck study, low toll-like receptor 4 (TLR4) expression was associated with a decreased risk of carotid atherosclerosis, as well as less inflammatory response to gram-negative pathogens, and high susceptibility to bacterial infections [101]. This is explained by the role of TLR4 as a promoter of NF-κB activation and inflammatory cytokine production. Therefore, genes may play an important role in the progression of atherosclerosis, even modulating the response of our organism to cardiovascular risk factors.

The pressing question is whether anti-inflammatory drugs are ready to be used in clinical practice. For the reasons explained above, we believe that colchicine specifically must have a role in cardiovascular therapy in secondary prevention. Given the wealth of evidence accumulated with lipid-lowering and antiplatelet drugs, these drugs must be the main therapy for atherosclerosis. Furthermore, lipid-lowering drugs also decrease the inflammatory response [62]. Therefore, in patients in whom we may presume that current therapies for atherosclerosis are going to largely reduce the cardiovascular risk, it is probably not worth adding colchicine. However, colchicine could be considered, for instance, in patients who are far from meeting current LDL targets. Patients with evidence of progression on atherosclerosis despite optimal therapy, and perhaps those with an enhanced inflammatory status could get additional benefit from this drug (Figure 3). Measuring residual inflammation is still an open question, and a combination of biomarkers and imaging may provide the optimal results. At present, it is likely that CRP plasma levels provide the best results, although this must be confirmed, as the colchicine trials did not define high inflammatory status as an inclusion criterion.

## 6. Conclusions

Atherosclerosis is a chronic disease with a clear link to inflammation. In recent decades, great improvements have been seen in the outcomes of cardiovascular patients. Despite the lack of cardiovascular benefit of classic anti-inflammatory drugs, recent work has demonstrated that other anti-inflammatory therapies such as canakinumab or colchicine, which work through different mechanisms of action, improve cardiovascular outcomes. This fact opens the possibility of a paradigm shift in secondary prevention of atherosclerosis. Even though antiplatelet drugs and the control of cardiovascular risk factors continue to be the cornerstone of therapy for atherosclerotic disease, we could consider the use of colchicine in some special cases, such as patients with suboptimal treatment or with proven clinical instability. In addition, it may be beneficial to put additional effort into understanding the residual inflammatory level of our patients to refine their therapeutic approach. Novel techniques may lead us to the discovery of new genetic and plasma biomarkers that may enhance our ability to identify populations with high inflammation at high risk of developing adverse outcomes. In this way, we will be able to use anti-inflammatory drugs in these subjects to effectively decrease the incidence of future cardiovascular events.

## Figures and Tables

**Figure 1 jcm-10-02835-f001:**
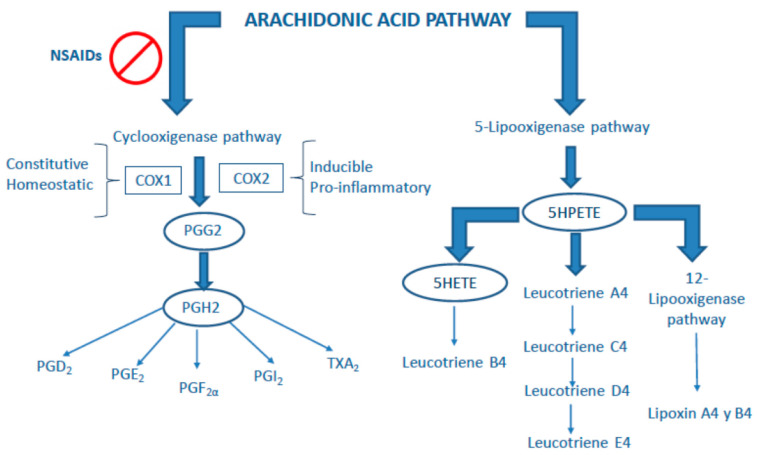
The arachidonic acid pathway. Abbreviations: COX: cyclooxygenase; NSAIDs: non-steroidal anti-inflammatory drugs; PG: prostaglandin; TX: thromboxane; HPETE: hydroperoxyeicosatetraenoic acid; HETE: hydroxyeicosatetraenoic acid.

**Figure 2 jcm-10-02835-f002:**
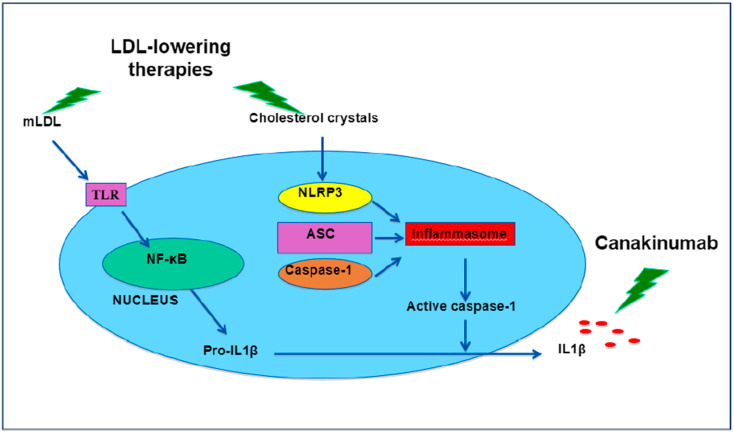
Relationship between lipids and IL1β activation. Lipids may induce the expression of pro-IL1β and its transformation into IL-1β. In this way, lipid-lowering therapy may decrease IL-1β expression. Canakinumab works on the same pathway blocking the effect of IL-1β, instead of modulating its expression. Reproduced from reference [83], with permission. Abbreviations: ASC: adaptor protein; IL-1β: interleukin 1β; m-LDL: modified LDL; NF-kB: nuclear factor kB; NLRP3: Nod-like receptor protein 3; LDL: low-density lipoprotein; TLR: toll-like receptors.

**Figure 3 jcm-10-02835-f003:**
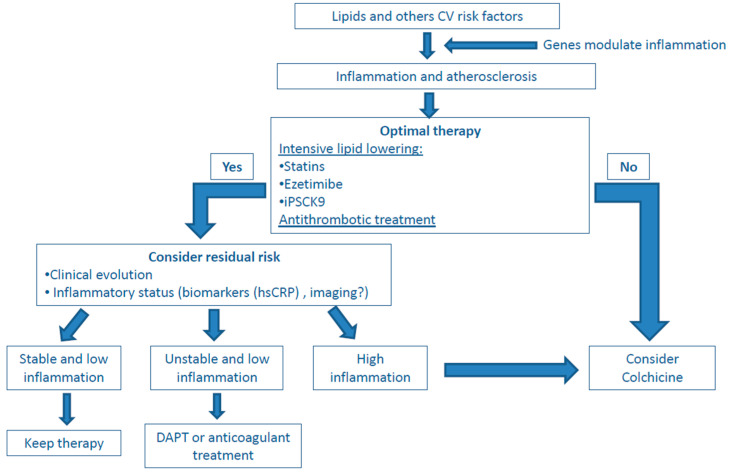
A potential approach to atherosclerosis and inflammation. Abbreviations: CV: cardiovascular; DAPT: double antiplatelet treatment; hsCRP: highly sensitive C-reactive protein; iPSCK9: Proprotein convertase subtilisin/kexin type-9 inhibitors.

**Table 1 jcm-10-02835-t001:** Summary of relevant clinical trials about canakinumab, methotrexate and colchicine.

Study	CANTOS [81]	CIRT [89]	LODOCO1 [90]	LODOCO2 [91]	COLCOT [92]	COPS [93]
**Trial population**						
CV previous disease	Previous MI	Previous MI or multivessel CD	Stable CAD	Stable CAD	MI 30 days before	ACS
High CRP mg/L	>2	No	No	No	No	No
Others		DM or metabolic syndrome			Complete percutaneous revascularization	Evidence of CAD
**Trial design**						
Participating centers	Multicenter	Multicenter	Single-center	Multicenter	Multicenter	Multicenter
Design	Double blind	Double blind	Observer blind	Double blind	Double blind	Double blind
Study drug and dose	Canakinumab 50;150;300 mg/3 months	Methotrexate 15–20 mg/weekly	Colchicine 0.5 mg/day	Colchicine 0.5 mg/day	Colchicine 0.5 mg/day	Colchicine 0.5 mg/day (twice daily first month)
Follow-up (years)	3.7	2.3	3	2.3	1.9	1
Sample size	10,061	4786	532	5522	4745	795
**Characteristics of the participants**						
Age yr	61	66	66.5	66	60.6	59.8
Male %	72.5	81	89	84.7	80.8	79.5
DM %	40	68	30.5	18.2	20.2	19
Statin %	93.4	85.9	95	94	99	98.5
Anti-thrombotic treatment	95.1	86.4	93	90.2	99	97.8
Median LDL cholesterol mg/dl	82.4	68	NR	NR	NR	NR
Median CRP mg/L	4.2	1.6	NR	NR	4.28 (only 207 patients)	NR
**End points**						
Non-fatal MI, stroke or CV death	Positive 150 mg HR 0.85 (0.74–0.98)	Negative HR 1.01 (0.82–1.25)	-	Positive HR 0.69 (0.57–0.83) *	Positive HR 0.77 (0.61–0.96) **	Negative HR 0.65 (0.38–1.09) ***
All-cause mortality	NS	NS	-	NS	NS	HR 8.20 (1.03–65.61)
ACS, out-of-hospital cardiac arrest, or non-cardioembolic ischemic stroke	-	-	Positive HR 0.33 (0.18–0.59)	-	-	-
NNT (patients)	156	-	11	91	62	-
**Adverse events in active drug**	More infection, neutropenia, thrombocytopenia	Higher liver enzyme levels, leukopenia and non-basal cell skin cancers	Intestinal intolerance	Less gout and more myalgia	More gastrointestinal events and pneumonia	NS
**Cost per year** **€**	45,957	22.8	43.2	43.2	43.2	46.8

Abbreviations: ACS, Acute coronary syndrome; CAD, coronary artery disease; CRP, C-reactive protein; CV, cardiovascular; DM, diabetes mellitus; HR, hazard ratio; MI, myocardial infarction; NNT, number needed to treat; NR, not reported; NS, not significant; LDL: low-density lipoprotein. * and ischemia-driven coronary revascularization. ** and resuscitated cardiac arrest or urgent hospitalization for angina leading to coronary revascularization. *** and ACS (instead of non-fatal MI), ischemia-driven (unplanned) urgent revascularization.

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
