# Peer review of "Anti-Inflammatory Drugs in Patients with Ischemic Heart Disease"

_jcm, 2021, doi:10.3390/jcm10132835_

Round 1

Reviewer 1 Report

The authors AM PelloLázaro, LM Blanco-Colio, JA FrancoPeláez and J Tuñón have prepared a very interesting review entitled: Anti-inflammatory drugs in patients with ischemic heart disease. They summarize up-to-date knowledge in the field with nearly 100 references, some very recent from 2020-2021 but there are unfortunately 2 lists in duplicate at the end of the article, and we didn’t have the time to check the consistency of these two. On the other side, there are clearly problems in the first list of them, with duplicates such as reference 29 and 39.

We understand that MCP-1 is the field of expertise of these authors, but there have been also recent interesting papers on MCP-1 worth commenting, next to reporting only their publications. We think e.g. about MK Georgakis et al in Arterioscler Thromb Vasc Biol. 2021;41:00–00. DOI: 10.1161/ATVBAHA.121.316091 or the extensive meta-analysis including 7 cohort studies involving 21,401 individuals published in 2000 in JAMA Cardiol. doi:10.1001/jamacardio.2020.5392.

About canakinumab, to our knowledge, and as made available on www.ema.europa.eu website in a withdrawal report, the benefit-risk balance is currently negative for secondary prevention of MACE in patients after a myocardial infarction and not approved due to data available only from one pivotal study not statistically compelling with regards to the strength of statistical evidence with only a 15% risk reduction for MACE in the overall group of patients but only 9% in the key subgroup of patients on baseline statin therapy. The clinical relevance of efficacy was questionable with on the other side increased risks for fatal infections and sepsis. So, when saying on line 243 “avoided the routine use of canakinumab”, this is felt like an understatement.

We agree with this review that colchicine might be the best anti-inflammatory drug to use. However, we do not understand the reasoning behind figure 3 and the proposal starting at line 380:  “colchicine could be used in patients with evidence of progression on atherosclerosis despite optimal therapy…those with an enhanced inflammatory status…combination of biomarkers and imaging” since the published trials such as LoDoCo-2 showed a 31% reduction in the primary endpoint without such imaging/biomarkers selection in +5,500 patients with chronic coronary syndrome (a better term to use than stable ischemic heart disease) randomized to receive colchicine or placebo. So, who should not get it… ACE inhibitors have been given to all for less than this effect?

Minor comments

15 missing space before ‘In recent years’

36 double space between ‘of’ and ‘Low density lipoprotein’

106 Lp-PLA2 does not affect the incidence of CV events because it plays a role as a biomarker

112 missing point after ‘[48,49]’

124 The number of the paragraph is still ‘1’, like all the other ones.

139-142 repetition from the introduction part : need to delete?

204 add the number of patient exposed to treatment in the meta-analysis (>2.7million) (comparison to PRECISION TRIAL)

275, table 1 : the font size is too small. The content isn’t easily readable.

289 double space between ‘outcomes’ and ‘[46,86]’ and missing point after ‘[46,86]’.

Author Response

ANSWERS TO REVIEWER 1:

First of all, we would like to thank the editor and the reviewers for their comments that have helped us to improve our manuscript.  We copy the reviewer’s comments (bold) followed by our answers (italic).

 The authors AM PelloLázaro, LM Blanco-Colio, JA FrancoPeláez and J Tuñón have prepared a very interesting review entitled: Anti-inflammatory drugs in patients with ischemic heart disease. They summarize up-to-date knowledge in the field with nearly 100 references, some very recent from 2020-2021 but there are unfortunately 2 lists in duplicate at the end of the article, and we didn’t have the time to check the consistency of these two. On the other side, there are clearly problems in the first list of them, with duplicates such as reference 29 and 39.

Thank you very much for your comment. You are right. We have eliminated the duplicated list and we have corrected the duplicated reference. We apologize for this mistake. We have also made several modifications in the bibliography using the *Track Changes* function.

 We understand that MCP-1 is the field of expertise of these authors, but there have been also recent interesting papers on MCP-1 worth commenting, next to reporting only their publications. We think e.g. about MK Georgakis et al in Arterioscler Thromb Vasc Biol. 2021;41:00–00. DOI: 10.1161/ATVBAHA.121.316091 or the extensive meta-analysis including 7 cohort studies involving 21,401 individuals published in 2000 in JAMA Cardiol. doi:10.1001/jamacardio.2020.5392.

We really appreciate your comment. Effectively, these very recent works have a good methodology and we have added these references to the new version of the manuscript. We believe that these studies provide relevant information to the paper.

 About canakinumab, to our knowledge, and as made available on www.ema.europa.eu website in a withdrawal report, the benefit-risk balance is currently negative for secondary prevention of MACE in patients after a myocardial infarction and not approved due to data available only from one pivotal study not statistically compelling with regards to the strength of statistical evidence with only a 15% risk reduction for MACE in the overall group of patients but only 9% in the key subgroup of patients on baseline statin therapy. The clinical relevance of efficacy was questionable with on the other side increased risks for fatal infections and sepsis. So, when saying on line 243 “avoided the routine use of canakinumab”, this is felt like an understatement.

Thank you for your comment. We agree with you on the great relevance of the EMA’s report, advising against the use of canakinumab in secondary prevention of patients with previous myocardial infarction. We mention this statement it in the revised version of the manuscript, and we have added its reference in the bibliography.

 We agree with this review that colchicine might be the best anti-inflammatory drug to use. However, we do not understand the reasoning behind figure 3 and the proposal starting at line 380:  “colchicine could be used in patients with evidence of progression on atherosclerosis despite optimal therapy…those with an enhanced inflammatory status…combination of biomarkers and imaging” since the published trials such as LoDoCo-2 showed a 31% reduction in the primary endpoint without such imaging/biomarkers selection in +5,500 patients with chronic coronary syndrome (a better term to use than stable ischemic heart disease) randomized to receive colchicine or placebo. So, who should not get it… ACE inhibitors have been given to all for less than this effect?

Thank you for this highly relevant comment. Perhaps the most important part of this review. As you know, we are using ACEi in a large number of patients due to their broad CV benefits: in addition to their anti-inflammatory effects they lower blood pressure, reduce LV remodeling after a myocardial infarction, and have been said to reduce thrombosis, …

However, colchicine is presumed to get a clinical benefit mainly through its anti-inflammatory effects. Although all patients on secondary prevention are theoretically at very high risk, they really represent a broad spectrum of risk. In this way, it is possible that after an isolated ACS some patients may be very well controlled with the traditional drugs, mainly if they reach the LDL target of 55 mg/dl, advised by ESC guidelines. Probably, adding one more drug may complicate adherence to therapy and provide no great additional benefit. In this regard, LDL levels were not reported in colchicine trials, but given the timing of these trials, we presume that most patients were not below this limit. Given that intensive lipid-lowering therapy further lowers the CV risk and that a part of this effect could be due to the anti-inflammatory effect of lipid reduction…can we anticipate that adding colchicine could further reduce the risk in patients with LDL <55 mg/dl? As we said, LDL targets advised by clinical guidelines are much strict now than at the moment these trials were performed, and we may expect that most of our patients have lower LDL levels currently than in the past. Then, we believe that a broad, non-selective use of colchicine in all patients on secondary prevention is not justified.

In the new version of the paper we have changed the text, emphasizing why we believe that colchicine should not be used in all patients on secondary prevention on a routine basis, and also adding patients who are far from meeting the current LDL targets (due to drug intolerance, or to restricted access to new therapies, such as PCSK9 inhibitors) as candidates for colchicine therapy. We have also modified Figure 3 according to these changes. We thank you for this comment, because we believe it has stimulated us to improve the manuscript

 Minor comments

15 missing space before ‘In recent years’.

You are right. We have fixed it

36 double space between ‘of’ and ‘Low density lipoprotein’.

You are right. We have corrected it

106 Lp-PLA2 does not affect the incidence of CV events because it plays a role as a biomarker.

We are not completely sure about the exact meaning. The text is:

“However, Lp-PLA2 is not a fully validated biomarker for clinical practice. In addition, Lp-PLA2 inhibition was not shown to affect the incidence of cardiovascular events”.

What we say is that, at present, Lp-PLA2 is not a validated biomarker. After this sentence, we add that Lp-PLA2 inhibition has not demonstrated clinical benefits. However, this is just additional information as, theoretically, the ability of Lp-PLA2 as a biomarker could be independent from the clinical effect obtained through its blockade. As we see that this may confounding, we have decided to delete it from the paper.

112 missing point after ‘[48,49]’

Done.

124 The number of the paragraph is still ‘1’, like all the other ones.

We have corrected it.

139-142 repetition from the introduction part: need to delete?

Thank you for your comment. We have deleted the duplicated text in the revised manuscript.

204 add the number of patient exposed to treatment in the meta-analysis (>2.7million) (comparison to PRECISION TRIAL).

 We have added it in the new version of the manuscript.

275, table 1: the font size is too small. The content isn’t easily readable.

We have increased font size and the table is now easier to red. However, this has changed the final shape. If the Editorial Board believes that it should be different, please let us know.

289 double space between ‘outcomes’ and ‘[46,86]’ and missing point after ‘[46,86]’. We have corrected it.

We want to thank you very much again for your suggestions and help.

Reviewer 2 Report

The article presents a well-organized review about the antiinflammatory drugs in atherosclerosis pahophysiology  management. Irt is well written, coincice, well referenced. the table about the trials is a little be busy but the figures are clear and self explaining.

As only very minor criticism in the paragraph 1.1, it would be good to spend 2-3 lines to underscore that atherosclerosis is not only an inflammatory disease but involves also the type of flow and the correlation between flow and coronary wall (helical flow, computational fluid dynamic studies, etc)

Author Response

ANSWERS TO REVIEWER 2:

 First of all, we would like to thank the editor and the reviewers for their comments that have helped us to improve our manuscript.  We copy the reviewer’s comments (bold) followed by our answers (italic).  

The article presents a well-organized review about the antiinflammatory drugs in atherosclerosis pahophysiology  management. Irt is well written, coincice, well referenced. the table about the trials is a little be busy but the figures are clear and self explaining.

Thank you for your comment. We believe that this highly relevant topic may be of interest for the readers due to the potential use of these drugs in the routine clinical practice.  

 As only very minor criticism in the paragraph 1.1, it would be good to spend 2-3 lines to underscore that atherosclerosis is not only an inflammatory disease but involves also the type of flow and the correlation between flow and coronary wall (helical flow, computational fluid dynamic studies, etc)

Thank you. We have changed the beginning of 1.1 paragraph according to this suggestion. 

We want to thank you very much again for your suggestions and help.

Reviewer 3 Report

The adverse effects of NSAIDs on cardiovascular events are well documented. However, the unlimited availability of this group of drugs makes the topic of the reviewed manuscript important and up-to-date.

The manuscript under review is an interestingly written review combining the theoretical basis of the role of inflammation in the pathogenesis of atherosclerosis with the presentation of the results of clinical studies on the effects of anti-inflammatory drugs in patients with coronary heart disease. The authors briefly discuss the pleiotropic anti-inflammatory effects of such drugs used in the therapy of the cardiovascular system as ACE-I and statins. In this group of drugs, they also position acetylsalicylic acid. Then they discuss the influence of currently used non-steroidal anti-inflammatory drugs on the course of coronary artery disease. At this point, it is worth considering the results of studies that documented the adverse effect of indomethacin in patients with coronary heart disease, published in the 1980s.

In order to better present the negative impact of NSAIDs on the course of coronary artery disease, I propose, similarly to colchicine and canakinumab, to present the results of meta-analyzes in the form of a table taking into account the meta-analysis by M Bally at al (M. Bally et al. Risk of acute myocardial infarction with NSAIDs in real world use: Bayesian meta-analysis of individual patient data. BMJ 2017;357:j1909)

In the further part of the manuscript, the authors present the current state of knowledge on the beneficial effects of colchicine and canakinumab in patients with coronary artery disease, having a mechanism of action different from that of NSAIDs.

Although the beneficial effects of colchicine use did not depend on inflammatory activation, the authors suggest that the group of patients with elevated inflammatory parameters may benefit most from colchicine therapy. They propose an algorithm of pharmacotherapy of patients with coronary artery disease taking into account inflammatory parameters (Table 3).

Undoubtedly, the results of the presented research are promising, although the final conclusions should be more subdued.

Minor revision:

Row 11 (and in other section): no ….”anti-atherosclerotic drugs”  better will be: “some drugs used in therapy for atherosclerotic diseases such as”

Row 11 (and in other section): please change aspirin on acetylsalicylic acid

Row 17 and in other section: Specifically, colchicine is a candidate for use in clinical practice given its safety, low price, and the considerable amount of data supporting its use.

I propose a more cautious wording: “Potentially, colchicine could be a candidate for use in clinical practice given its safety and low price, although the results of temporary studies require confirmation in large randomized clinical trials”.

Row 275: “new anti-inflammatory drugs” - colchicine is not a new anti-inflammatory drug, instead of "new inflammatory drugs, you can use the names of the investigated drugs

Author Response

ANSWERS TO REVIEWER 3:

 First of all, we would like to thank the editor and the reviewers for their comments that have helped us to improve our manuscript.  We copy the reviewer’s comments (bold) followed by our answers (italic).  

The adverse effects of NSAIDs on cardiovascular events are well documented. However, the unlimited availability of this group of drugs makes the topic of the reviewed manuscript important and up-to-date.

The manuscript under review is an interestingly written review combining the theoretical basis of the role of inflammation in the pathogenesis of atherosclerosis with the presentation of the results of clinical studies on the effects of anti-inflammatory drugs in patients with coronary heart disease. The authors briefly discuss the pleiotropic anti-inflammatory effects of such drugs used in the therapy of the cardiovascular system as ACE-I and statins. In this group of drugs, they also position acetylsalicylic acid. Then they discuss the influence of currently used non-steroidal anti-inflammatory drugs on the course of coronary artery disease. At this point, it is worth considering the results of studies that documented the adverse effect of indomethacin in patients with coronary heart disease, published in the 1980s.

We really appreciate your comment. We fully agree with your mention about indomethacin. We have reviewed the literature and, following your recommendation, we have included in the revised manuscript some relevant references about this drug.

In order to better present the negative impact of NSAIDs on the course of coronary artery disease, I propose, similarly to colchicine and canakinumab, to present the results of meta-analyzes in the form of a table taking into account the meta-analysis by M Bally at al (M. Bally et al. Risk of acute myocardial infarction with NSAIDs in real world use: Bayesian meta-analysis of individual patient data. BMJ 2017;357:j1909).

Thank you for your comment. We have included the meta-analysis by Bally et al in the new version of the manuscript.  We believe that a table including the most relevant data on NSAIDs studies is really attractive. Nevertheless, these studies have heterogeneous methodologies, and it would be very difficult to unify them. In addition, the existence of adverse cardiovascular effects with NSAIDs is broadly accepted at present, and it is not under discussion. Then, we believe that adding more data on this specific issue may be beyond the scope of this review. Instead of this, we have chosen to focus on the results of canakinumab and colchicine, as these are recent data that may be helpful to make clinical decisions on the potential use of these drugs in patients with coronary artery disease.

In the further part of the manuscript, the authors present the current state of knowledge on the beneficial effects of colchicine and canakinumab in patients with coronary artery disease, having a mechanism of action different from that of NSAIDs.

Although the beneficial effects of colchicine use did not depend on inflammatory activation, the authors suggest that the group of patients with elevated inflammatory parameters may benefit most from colchicine therapy. They propose an algorithm of pharmacotherapy of patients with coronary artery disease taking into account inflammatory parameters (Table 3).

Undoubtedly, the results of the presented research are promising, although the final conclusions should be more subdued.

To tell the truth, discussing the need to measure the inflammatory status is a complicated issue, given that in colchicine trials it was not assessed. Also, it is clear that we do not have currently a widely accepted method to assess CV inflammation. In order to decrease the weight of this point both in the conclusion and in the algorithm, we now refer to patients with clinical instability (repeated CV events under adequate therapy) and suboptimal therapy as potential candidates for anti-inflammatory therapy, and we have moved the issue of assessing the inflammatory status to a less remarkable second level. These changes have been also included in Figure 3. We thank you by this suggestion, as we believe that it has improved the quality manuscript. 

 Minor revision:

Row 11 (and in other section): no….”anti-atherosclerotic drugs” better will be: “some drugs used in therapy for atherosclerotic diseases such as”

We have corrected it

Row 11 (and in other section): please change aspirin on acetylsalicylic acid.

Done

 Row 17 and in other section: Specifically, colchicine is a candidate for use in clinical practice given its safety, low price, and the considerable amount of data supporting its use.

 I propose a more cautious wording: “Potentially, colchicine could be a candidate for use in clinical practice given its safety and low price, although the results of temporary studies require confirmation in large randomized clinical trials”.

Thank you. In the new version of the manuscript we use a more cautious wording, as you suggest.

 Row 275: “new anti-inflammatory drugs” - colchicine is not a new anti-inflammatory drug, instead of "new inflammatory drugs, you can use the names of the investigated drugs.

 You are right. We have changed the title of Table 1, as you suggest.

We want to thank you very much again for your suggestions and help.

Round 2

Reviewer 1 Report

line 306: major, not mayor

this paper is much better now